# DEEP COGNITION: TOWARDS A MORE TRANSPARENT AND INTERACTIVE RESEARCH AGENT

## ABSTRACT

Despite advances in large language models(LLMs), current systems for deep research are limited by an asynchronous, "Input-Wait-Output" interaction paradigm. This model creates a critical disconnect between human intent and AI execution, leading to error propagation and an inability to dynamically course-correct during complex problem-solving. We introduce Deep Cognition, a system designed to enable this paradigm through three technical pillars: transparent and interruptible AI reasoning, fine-grained bidirectional dialogue, and a shared cognitive context. At the core of our system is a multi-agent collaboration framework driven by a dynamic Plan-Search-Report workflow. This architecture continuously integrates interaction data (information) (e.g., dialogue trajectories and retrieved evidence) into an evidence-driven iterative report construction process. By employing selective context retention to filter unutilized information, our system mitigates error cascades and allows the AI to adapt its reasoning pathways based on the user's implicit focus. We conduct a comprehensive user study on challenging deep research tasks to evaluate the efficacy of our system. Results show that our approach significantly enhances the user experience, yielding improvements of up to 29.2% in Fine-Grained Interaction and 27.7% in Ease of Collaboration compared to a competitive baseline. Most notably, our system demonstrates a 31.8% to 50.0% improvement in overall task performance. These results highlight the critical importance of designing interactive AI systems that facilitate continuous human guidance and transparent reasoning, rather than merely responding to isolated commands.

## 1 INTRODUCTION

As artificial intelligence (AI) capabilities have advanced dramatically through large language models (LLMs) (Luo et al., 2024; Radford et al., 2018; 2021; Brown et al., 2020; 2024), the prevailing trajectory in AI development has emphasized scaling model parameters (Kaplan et al., 2020; Hoffmann et al., 2022; Wei et al., 2022), expanding training data (Yang et al., 2025; Meta AI, 2025), and refining architectures (DeepSeek-AI et al., 2025; MiniMax et al., 2025; Poli et al., 2024)—creating increasingly autonomous black boxes that assume minimal human input beyond simple prompting (Liu et al., 2023b; Kim et al., 2023), instruction (Kim et al., 2023) or decision-making (Yin, 2025). This pathway implicitly assumes that the ultimate form of artificial intelligence would require minimal human input. We contend that this assumption mischaracterizes the nature of intelligence itself. This paradigm positions humans as external operators who provide initial prompts and consume final outputs while remaining excluded from the cognitive process itself, treating human intelligence as merely an instructor rather than a collaborative partner. Consequently, a fundamental question emerges: ***How can we design an agentic framework that enables humans to effectively guide AI reasoning trajectories through strategic, real-time interventions?*** However, intelligence—whether human or artificial—is inherently interactive, contextual, and collaborative (Hutchins, 1995; Minsky, 1987; Woolley et al., 2010). The most sophisticated human thinking rarely occurs in isolation but emerges through dialogue, feedback, refinement, and the integration of diverse perspectives. Consider the nature of breakthrough scientific discoveries or complex problem-solving scenarios: They invariably involve iterative cycles of hypothesis formation, testing, revision, and collaborative refinement. As AI systems approach advanced cognitive capabilities powered by inference-time scaling (OpenAI, 2024)—enabling thought-level communication where strategic human oversight can leverage vast AI execution power (Xia et al., 2025)—the need for meaningful interaction transforms and intensifies.

This is especially critical for extended AI tasks (Kwa et al., 2025) spanning hours to days, which fundamentally alter human-AI collaboration dynamics.

This transition is particularly evident in systems designed for Deep Research tasks (OpenAI, 2025c; Google, 2025; Perplexity AI, 2025; Zheng et al., 2025a)—complex, extended cognitive processes involving dynamic information retrieval, filter, understanding, analysis and synthesis. Current state-of-the-art research systems have pioneered capabilities for multi-step web browsing, data analysis, and report generation. However, these systems uniformly adopt an "Input-Wait-Output" interaction paradigm where users initiate a query, wait through an extended "Black Box" processing period (typically 5-30 minutes), and eventually receive a comprehensive result. This approach reflects the persistent assumption that interaction is merely a necessary cost rather than a source of value. Yet these systems fundamentally suffer from critical deficiencies: early errors (Cemri et al., 2025) compound without correction, systems cannot adapt to evolving requirements, domain expertise remains inaccessible at crucial moments, and opaque processing prevents human-AI collaboration.

These deficiencies stem from a fundamental misalignment: systems that minimize human involvement during processing cannot address problems that require adaptive guidance and expert intervention (Bainbridge, 1983). To address this fundamental challenge, we develop **deep cognition**—a systematic framework that transcends traditional automation by embedding real-time human expertise directly into AI reasoning processes for complex research tasks, guided by the following principles:

- *Transparency:* The system reveals its entire thinking process—from search strategies and query formulations to information evaluation and synthesis rationales—making AI cognition inspectable and editable at every stage. This transparency enables true thought-level interaction where humans can guide how AI thinks.
- *Fine-Grained Interaction:* Users can engage with any specific element of the AI's output—questioning particular claims, requesting elaboration on specific points, or changing the research focus.

These principles fundamentally transform deep research from conventional question-and-answer exchanges into cognitive collaboration (see Appendix **??**) — what we term **cognitive oversight**. Rather than relegating humans to the role of passive tool operators, this framework establishes a synergistic reasoning process that harnesses the complementary strengths of human expertise and AI capabilities while mitigating their respective limitations. Through cognitive oversight, we move beyond the traditional paradigm of human-AI interaction toward a new form of augmented intelligence where strategic human insight and AI computational power merge into a unified cognitive system.

Through extensive experiments with real expert interactions, we demonstrate that deep cognition achieves substantial improvements or competitive over strongest baseline across all evaluation dimensions: Transparency (+20.0%), Fine-Grained Interaction (+29.2%), Real-Time Intervention (+18.5%), Ease of Collaboration (+27.7%), Results-Worth-Effort (+8.8%), and Interruptibility (+20.7%). Our contributions are summarized as follows:

- **Agentic Multi-Agent Workflow**: We developed an anti-degradation workflow that co-evolves with stronger base models and integrates professional sub-agents.
- **Comprehensive Evaluation Framework**: We establish a complete evaluation framework, including 15 metrics specifically designed for assessing the effectiveness of cognitive oversight in deep research scenarios.
- **Human Fine-Grained Oversight**: We operationalize the cognitive oversight paradigm into our multi-agent human-AI collaboration system designed for deep research tasks.

## 2 METHODOLOGY

### 2.1 SYSTEM ARCHITECTURE OVERVIEW

We propose a multi-agent collaborative deep research system designed to address the challenges of long-form report generation. The system supported by four key processes: Planning Agent, Clarification, Browsing Agent, and Writing Agent, with the capability for agents to solicit human

input at any stage of the cycle. The system workflow proceeds as follows: Initially, after user input, the **Proactive** **Clarification module** guides dialogue through structured questioning to precisely capture research intent and background information. After establishing research objectives, the system enters a **Plan-Search-Report** dynamic loop: within each cycle, network search queries are generated based on current planning status and delegated to the **Sub Browse-Agent Cluster**, which coordinates Sub-Agent groups to concurrently navigate and extract information from multiple web pages. During evidence collection, the **Writing Agent** continuously outputs intermediate reports, enabling dynamic user feedback. The workflow supports asynchronous human interruption at any stage, while agents can proactively evaluate whether the current report matches user requirements at the end of each round, seeking additional information to decide the next action if necessary. This design ensures the transparency of the research process while maintaining efficient automated information processing capabilities. The following subsections detail each core component and their technical implementation.

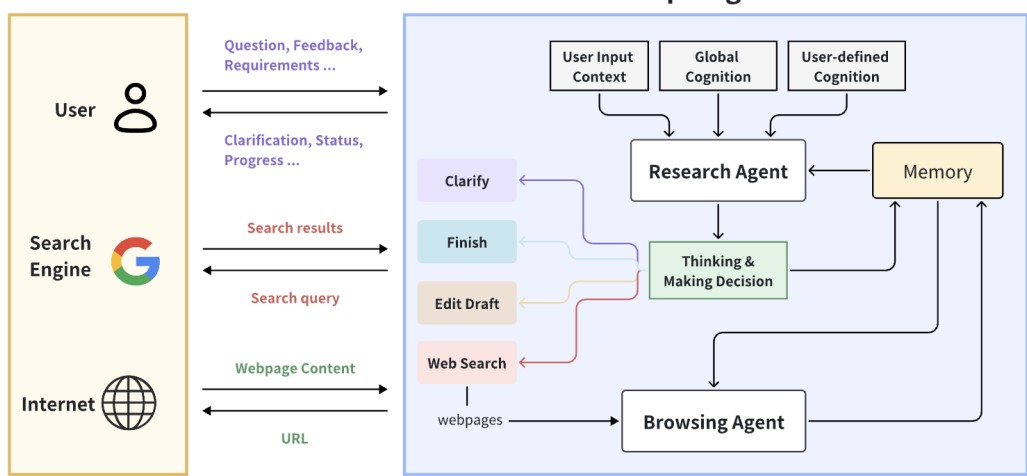

Figure 1: Deep cognition framework overview. This multi-agent research assistant system breaks down complex research questions and dynamically synthesizes information from multiple sources through iterative search, clarification, and user feedback.

## 2.2 MULTI-ROUND CLARIFICATION MECHANISM

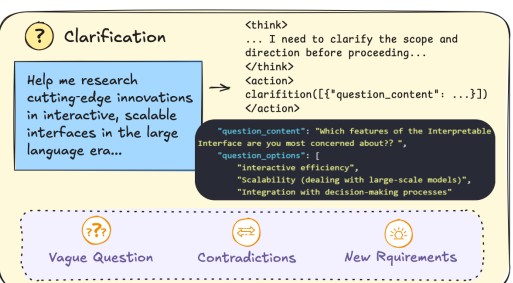

Existing deep research systems such as OpenAI DeepResearch (OpenAI, 2025b) and Gemini DeepResearch (Xu & Peng, 2025) typically conduct one-time question collection during the initial dialogue phase, but this approach neglects the dynamic clarification needs that emerge during the research process. Human researchers actively seek clarification for newly discovered points of confusion during exploration, and this timely feedback mechanism is crucial for research efficiency and quality.

Figure 2: Caption

We design an **option-driven progressive clarification framework** that transforms complex clarification questions into structured option questionnaires, rather than relying on traditional free-text input (detailed clarification prompts see appendix A.1). This mechanism supports triggering clarification processes at any stage of the research, providing continuous human supervision signals for subsequent information retrieval and report generation. **Proactive Clarification Trigger Mechanism:** The system employs a **prompt-based trigger mechanism** to identify moments requiring user clarification. Specifically, we design comprehensive scenario templates in the system prompt that guide the LLM

to recognize situations necessitating human input, including but not limited to: **Ambiguity Detection**: When the research question contains multiple interpretations or the scope is unclear. **Information Conflict**: When retrieved sources present contradictory claims or evidence. **Branch Decision Points**: When the research path encounters multiple viable directions requiring user preference. **Domain Expertise Gaps**: When the system encounters specialized terminology or domain-specific context beyond its knowledge.

To prevent over-interruption, the system can view all previous user interactions in the historical track. The LLM will review this history to avoid repetitive questions and ensure that each clarification request provides incremental value to the research process.

**Dynamic Option Generation:** Unlike systems that rely on predefined question templates, our framework employs. **dynamic option generation**. When a clarification need is identified, the system generates appropriate question. **Multiple-choice questions** with 3-5 options covering probable user intents. **Open-ended questions** for scenarios requiring free-form input. **Contextual explanations** to help users understand each option.

## 2.3 PROFESSIONAL AGENT CLUSTER

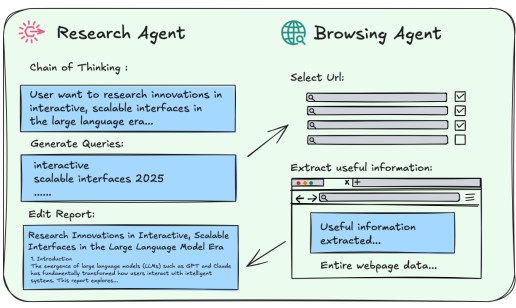

Figure 3: Caption

When processing large-scale web information retrieval tasks, we face two core challenges. First, the **information overload problem** arises as massive URLs and PDF documents exceed the effective processing range of a single model. Second, the **long-sequence degradation problem** manifests as existing large language models universally exhibit the "lost in the middle" (Liu et al., 2023a) phenomenon, struggling to effectively integrate scattered key information when processing long texts. Additionally, the inherent structural looseness and uneven information density of web content further exacerbate the complexity of information extraction. To address these challenges, we propose a distributed Sub-Browse Agent cluster architecture that achieves efficient information extraction through a systematic workflow. The main Research Agent first queries the Serper API to retrieve the top-20 candidate URLs for each search query, then strategically distributes these resources among specialized Sub-Agent instances. Each Sub-Agent operates within an isolated contextual environment to avoid cross-domain information interference.

For content processing, Sub-Agents employ adaptive chunking strategies to handle documents of varying lengths. Standard web pages are processed using fixed-size chunking with overlapping windows, while exceptionally long documents trigger an autonomous pagination decision mechanism where the Browse Agent evaluates content density and relevance to determine whether to continue processing subsequent sections. Upon completion of analysis, each Sub-Agent submits structured findings to the main Agent with three components: **Excerpts**, **Useful** and **Reasoning**. This architecture effectively distributes computational load, enables specialized processing optimization, and significantly improves both efficiency and accuracy in large-scale web information retrieval tasks.

The system utilizes a hierarchical, modular design to manage long-term research planning. We design a research agent to propose the research plan and autonomously determine the next action base on the current research state. This multi agent modular transfer isolates task-specific logic (e.g., research planning, web browsing, report generation), thereby preventing cross-module context interference. The research agent logs all completed events as a to-do list, this to-do list verified the current research state whether align with user goal.

We define the Research Agent (detailed prompts in appendix A.3) as a professional research scientist and strictly define the system's capability boundaries to enable the agent to plan highly feasible to-do lists (specifically capable of searching, analyzing, and writing, but not programming or deploying models). Furthermore, we provide the agent with three distinct few-shot example types (covering literature review, technical proposal, and precise retrieval), each including both correct and incorrect instances. These examples differentiate strategies suitable for internal deliberation, external informa-

tion seeking, and precise factual comparison, guiding the agent to generate the most accurate plans, detailed prompts in appendix A.2.

## 2.4 INTERMEDIATE REPORTS THROUGH WRITING AGENT

While existing deep research systems (LangChain, 2024; Roucher et al., 2025a) typically follow a sequential collect-then-generate paradigm, we propose an **evidence-driven iterative report construction strategy**. We deployed a specially fine-tuned Writing Agent capable of generating structured intermediate reports even when evidence collection remains ongoing (detailed prompts in appendix A.4).

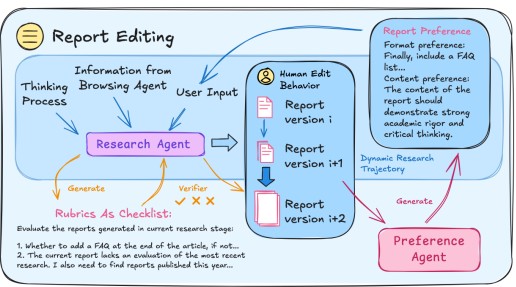

Figure 4: Caption

The system dynamically generates or adjusts hierarchical research plans at the beginning of each information collection cycle, with these plans serving as report outlines to guide the current cycle's writing tasks. This progressive synthesis approach delivers two key advantages: through **reasoning space construction**, it provides the model with a dedicated arena for deep reasoning and analysis during iterative optimization of multiple report versions; through **selective context retention**, the system preserves only the browsing results that have been incorporated into the current report, while directly removing unutilized evidence from subsequent processing contexts. This parallel evidence acquisition and report construction paradigm breaks through the limitations of traditional batch processing approaches, enabling continuous knowledge synthesis processes.

## 3 HUMAN-AI CO-RESEARCH MECHANISM

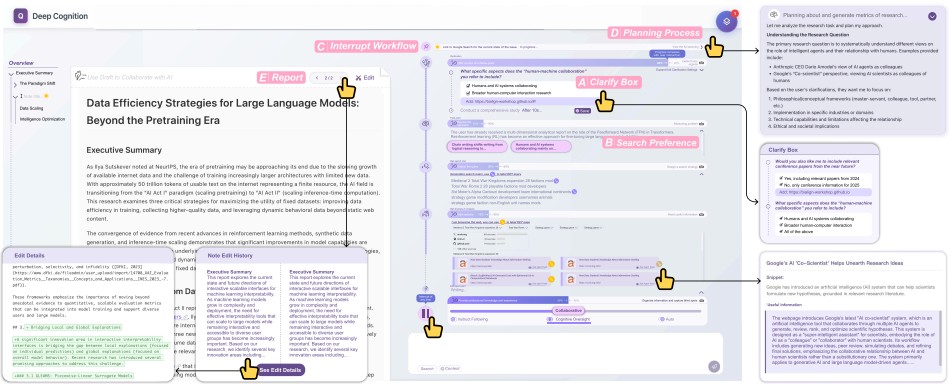

Figure 5: Deep cognition interface design showcasing key interactive features: (A) Research scope clarification to refine vague queries, (B) Click to open the important URL, (C) Multi-agent Workflow Visualization, (D) Transparent display of reasoning, research processes, and interactive query refinement, and (E) Report revision. The 👆 icon stands for clickable interface elements.

Deep cognition supports real-time human–AI collaboration. It is designed for open-ended, multi-hop retrieval and exploratory analysis. It enables users to iteratively expand the initial question and produce a synthesized write-up. Following principles of cognitive oversight, we designed the following features for our deep cognition system, with interfaces presented in Figure 10. The interface supports multiple modes of human–AI collaboration: Clarification (left): The system generates clarification questions to help users specify their focus. Interrupt (bottom-left): Users can intervene during the system's ongoing retrieval or reasoning process, halting unsatisfactory results and redirecting the search toward more relevant information. Planning (right): The system synthesizes retrieved evidence into a structured research plan. **Transparent Research Process:** The interface

make the system's decision-making process visible and comprehensible to users. Search strategy explainability is achieved by directly displaying the reasoning process and query terms generated by the model, making information retrieval interpretable. The editor area on the left of Figure 10 displays the evolving research document with proper formatting. All findings are properly linked to their original sources, enabling users to trace source materials. **Real-Time Intervention** We implement a "Pause" feature, allowing users to interrupt the system at critical junctures in the research process. This intervention capability enables users to actively shape the research trajectory based on emerging insights or changing objectives.

## 4 EXPERIMENTS

### 4.1 METRICS DESIGN

We defined five key dimensions for evaluating the quality of generated reports. Each metric is rigorously assessed using a 5-point Likert scale. For report quality, we focus on organization, coverage, depth, relevance, usefulness, and innovation. For interaction dimensions, we focus on willingness to use, usability, transparency, interruptibility, granular interaction, informativeness, ease of collaboration, cost-effectiveness, real-time intervention, and usefulness.

| Metric | Description | Metric | Description |
|---|---|---|---|
| Organization | Evaluate whether the article demonstrates sound organization and logical structure. An acceptable response should: (1) Exhibit clear structure by organizing relevant points into a coherent logical sequence. (2) Maintain coherence without any contradictions or unnecessary repetition. | Intention to Use | Measures user intention and propensity for continued engagement with the system based on perceived value and satisfaction. |
| Cutting-Edge | Assess whether the article demonstrates comprehensive coverage of existing literature by: (1) Effectively summarizing and conducting comparative analysis with previous research. (2) Timely incorporating the most recent and up-to-date research findings or information. | Usability | Evaluates the intuitive nature and accessibility of the system interface, including cognitive load and interaction efficiency. |
| | | Transparency | Assesses the interpretability and explainability of the model's decision-making processes and reasoning mechanisms. |
| Coverage | Provide comprehensive coverage of the identified areas of interest through: (1) Conducting thorough reviews. (2) Citing a broad range of representative scholarly works. (3) Incorporating the most current and time-sensitive information from various sources, rather than limiting the analysis to a small number of papers. | Interruptibility | Assesses the system's ability to tolerate pauses or context switches and to resume smoothly without loss of state or progress. |
| | | Fine-Grained Interaction | Evaluates the system's capacity to incorporate user feedback and enable precise, granular control over output generation. |
| | | Inspiration | Assesses the system's ability to stimulate creative thinking and generate ideas or innovative approaches to problem-solving. |
| Depth | Assess the adequacy of information content provided in the article. Specifically, evaluate whether the article delivers sufficient relevant information with appropriate depth such that readers can achieve thorough understanding of each argument presented. | Ease of Collaboration | Measures the extent to which the system functions as an effective collaborative partner in knowledge work and decision-making processes. |
| | | Results-Worth-Effort | Evaluates whether users perceive the time and effort invested in system interaction as worthwhile and valuable relative to the outcomes achieved. |
| Relevance | Assess whether the response maintains topical relevance and preserves clear focus in order to deliver a useful response to the posed question. Specifically, the output should: (1) Sufficiently address the central elements of the original question and satisfy your informational requirements. (2) The response should exclude substantial amounts of tangential information unrelated to the original inquiry. | Real-Time Intervention | Measures the degree to which users can actively interrupt and steer the system's ongoing processes—e.g., pausing, editing, or re-prompting—to obtain desired outputs. |
| | | Helpfulness | Assesses the overall utility and practical value of the output in addressing user needs and facilitating problem-solving objectives. |

Figure 6: Evaluation Metrics for Report Quality Assessment

### 4.2 SYSTEM EXPERIMENTAL SETUP

We use claude-3.7-sonnet-thinking as an inference model for action selection and claude-4.0-sonnet for document authoring, and the browsing agent uses gpt-4.1-mini for processing large numbers of documents, with 0.6 used for both temperature. We used the Google TOP20 for web search to provide a realistic search environment for the Agent System. Each turn search generate 5 queries, and for 5 webpages for each query.

## 4.3 RESEARCH TASK SETUP

To addresses two limitations of static benchmarks, We perform a human evaluation to evaluate the real-world human experience during the human-AI interaction inspired by Lee et al. (2024). This method enables assessment of output quality that depends on interactive dynamics, which aligns with real-world usage scenarios. We develop a web application for users to interact with deep cognition in real time. We compare it with three competitive deep research baseline: Gemini Deep Research (Google, 2025), OpenAI Deep Research (OpenAI, 2025b;a;c) and Grok 3 DeeperSearch (xAI, 2025). Study 1 measuring report quality and the effectiveness of the interaction design. Study 2 testing whether users with higher or lower prior knowledge levels show differences in multi-hop retrieval task.

**Study 1** We recruited 13 participants with prior research experience. Before using the system, they were introduced to our evaluation metrics(see section 4.1) for deep cognition to ensure a shared understanding. Participants then evaluated both the quality of generated reports and the system's interactive behaviors on a 5-point Likert scale, supplemented by qualitative responses to open-ended interviews. Each participant proposed a research question from their own work, participants observed the model in real time as it retrieved information, reasoned through intermediate steps, and generated self-evaluations. They could not directly edit the final report but instead guided the process via interactive mechanisms such as interrupting outputs, injecting prior knowledge, inspecting sources, reviewing self-evaluations, suggesting new directions, giving feedback, or contributing personal documents. These interventions helped steer the model toward deeper analysis and more efficient retrieval, with the report finalized when the model itself chose to conclude.

**Study 2** To validate our hypothesis that experts with higher cognitive capabilities demonstrate enhanced collaboration with AI in transparent dialogue environments, we measured system performance through two comprehensive benchmarks. Given that our expert annotators are native Chinese speakers with domain expertise, we selected representative subsets for intensive interactive evaluation: 22 questions from browsecomp-ZH (Zhou et al., 2025) (top two from each of 11 categories) and the first 20 questions from xbench-deep research (Chen et al., 2025). Both sampling strategies ensure feasible human-AI collaborative assessment.

## 5 MAIN RESULT

### 5.1 EXPERT USER EVALUATION

As shown in Table 1, augmented through expert interaction, the deep cognition system demonstrated significant enhancements across six evaluated metrics, overall average improve 63%. Notably, the ORGANIZATION exhibits the greatest gain (+97%), followed by CUTTING-EDGE (+79%) and depth (+76%). Even the dimension with the smallest gain, helpfulness, showed a significant improvement of +42%. As the evaluation results in Table 2, the **alignment between expert rankings and user evaluations** validates our core hypothesis: **The system with enhanced interaction mechanisms consistently deliver output quality across six metrics.**

| Metric | DC (w/o Int). | DC. |
|---|---|---|
| Organization | **2.231** | **4.385 ↑ 97%** |
| Cutting-Edge | **2.538** | **4.538 ↑ 79%** |
| Coverage | 2.423 | 4.000 ↑ 65% |
| Depth | **2.231** | **3.923 ↑ 76%** |
| Relevance | 2.885 | 3.769 ↑ 31% |
| Helpfulness | 2.808 | 4.000 ↑ 42% |
| Overall Average | 2.519 | 4.103 ↑ 63% |

Table 1: Performance improvement of deep cognition over deep cognition without interaction. DC. indicates deep cognition, DC (non). indicates deep cognition without interaction.

Deep cognition dominates six of the seven metrics. It records the largest gains in Fine-Grained Interaction (+44.6%) and Cooperative (+43.0%), and is the only system to reach a perfect Transparency score (5.00, +25.0% over the strongest baseline). Overall, the results highlight deep cognition's superior transparency, controllability, and collaborative support. These quantitative results are further supported by users' qualitative feedback. Over 90% of participants agree or strongly agree that interaction with deep cognition improves report quality; 69% find it easy to use and 62% show a high willingness to use.

| Report Evaluation (1–5 Score) | | | | |
|---|---|---|---|---|
| **Metric** | **DC.** | **Gemini** | **OpenAI** | **Grok3** |
| Organization | **4.385**+1.8% | 4.308 | 3.769 | 3.385 |
| Cutting-Edge | **4.538**+3.5% | 4.385 | 3.769 | 3.538 |
| Coverage | 4.000-10.4% | **4.462** | 3.692 | 2.923 |
| Depth | 3.923-1.9% | **4.000** | 3.577 | 2.769 |
| Relevance | 3.769-18.3% | **4.615** | 4.077 | 3.615 |
| Helpfulness | **4.000**+0.0% | **4.000** | 3.615 | 2.692 |

| Interaction Evaluation (1–5 Score) | | | | |
|---|---|---|---|---|
| **Metric** | **DC.** | **Gemini** | **OpenAI** | **Grok 3** |
| Transparency | **5.00**+25.0% | 4.00 | 3.00 | 3.19 |
| Interruptibility | **4.35**+31.4% | 3.31 | 2.69 | 2.62 |
| Fine-Grained Interaction | **4.73**+44.6% | 3.27 | 2.88 | 2.19 |
| Real-Time Intervention | **4.69**+24.4% | 3.77 | 2.92 | 2.62 |
| Inspiration | **4.08**+0.0% | **4.08** | 3.42 | 3.19 |
| Ease of Collaboration | **4.62**+43.0% | 3.23 | 2.77 | 1.85 |
| Results-Worth-Effort | **4.52**+10.8% | 4.08 | 3.29 | 2.96 |

Table 2: User and expert evaluation results for AI research assistance systems. Left panel: User-generated evaluation scores on a 1-5 scale, where participants queried systems with their own research questions. Right panel: Scores (1–5 scale) for system-interaction evaluation metrics. Color coding indicates within-row performance rankings, and percentages show deep cognition's relative improvement over the strongest baseline system (Gemini). DC. indicates deep cognition.

## 5.2 BENCHMARK EVALUATION RESULTS

The results provide compelling evidence for our collaborative cognition framework. On browsecomp-ZH, the deep cognition system achieves 72.73% accuracy—dramatically outperforming all baselines (Gemini/OpenAI: 40.91%, Grok 3: 22.73%). Ablation studies show neither cognitive oversight alone (45.45%) nor interaction alone (40.91%) match their combination. On X-bench, our system achieves 65% accuracy, matching OpenAI while substantially outperforming Gemini (35%). Note that browsecomp-ZH was evaluated on June 22, 2025, and X-bench on September 25, 2025—temporal gaps may contribute to baseline performance variations due to API updates. The results consistently demonstrate that expert-AI collaboration requires both transparent reasoning and interactive guidance for effective performance across domains. Participants with deeper cognitive processing capabilities achieved significantly higher human-AI collaborative performance compared to those with surface-level cognitive approaches in transparent interaction paradigms, as measured by problem resolution accuracy.

| | DC (non cog). | DC (non int). | DC (cog+int). | Gemini | OpenAI | Grok 3 |
|---|---|---|---|---|---|---|
| Accuracy | 45.45% | 40.91% | 72.73% | 40.91% | 40.91% | 22.73% |

| | DC (cog+int). | Gemini | OpenAI |
|---|---|---|---|
| Accuracy | 65% | 35% | 65% |

Table 3: Accuracy comparison across benchmarks. Top: Browsecomp-ZH (22 questions). Bottom: X-bench deep research (first 20 questions). DC (non cog). = baseline with middle school-level participants (n=4); DC (non int). = autonomous system; DC (cog+int). = interactive condition with graduate-level participants (n=4).

## 5.3 IN-DEPTH ANALYSIS OF THE HUMAN STUDY: HUMAN HOLD DYNAMIC MENTAL MODELS THROUGHOUT COLLABORATION PROCESS

Enhancing transparency at the model's behavioral status can improve human-AI collaboration. Specifically, in complex, long-duration retrieval tasks, humans tend to delegate mechanical operations such as "browsing" and "summarizing" to AI, while preferring to collaborate with the model at decision points requiring higher-order thinking. We dive deeper into the human behavior pattern in the deep research process and provide design considerations of human-AI collaboration research system. As illustrated in case study(see Appendix E) and User Behavior Data Point (see Appendix C), our user study reveals a sophisticated pattern of collaborative engagement that varies systematically across six research phases. Users demonstrate **dynamic cooperation willingness**, transitioning between "hands-on" and "hands-off" modes based on task characteristics and their domain expertise. We detail these six phases below:

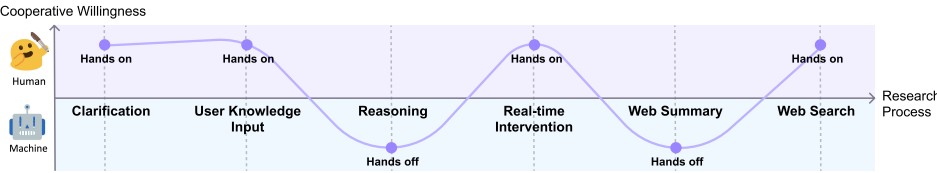

Figure 7: Changes in users' behavioral tendencies in the process of complex research tasks.

**Clarification (Hands-on)** The research process begins with intensive human-AI collaboration as users refine vague problem definitions. Users' initial research questions are typically too broad to cover all possible scenarios. **User Knowledge Input (Hands-on)** Users maintain high engagement when they possess specific domain knowledge or references that need integration. When users know specific references or attributes about an item, such as queries, paper links, websites, or personal opinions, they actively guide the AI to relevant media. **Reasoning (Hands-off)** Users seek to understand whether the model has correctly executed prescribed instructions and want transparency in decision-making processes. **Real-Time Intervention (Hands-on)** Cooperation peaks again during dynamic browsing tasks where users encounter pages or information sources that warrant detailed retrieval. **Web Summary (Hands-off)** During summarization tasks, users tend to trust in AI capability. Participants often need consolidated insights from multiple sources rather than single source summarization, leading them to allow extended autonomous operation. **Web Search (Hands-on)** The cycle concludes with renewed hands-on engagement for open-ended and subjective questions that require interpretation or subjective judgment.

This dynamic pattern demonstrates that effective human-AI collaboration is not uniform but adapts strategically to leverage the comparative advantages of human judgment and AI processing capabilities across different research phases. We illustrate this dynamic research task example to demonstrate authentic participant behavior.

## 6  RELATED WORK

**Human-AI Interaction**   AI agents White (2024); Feng et al. (2025) now support complex tasks through natural language interaction, better task understanding, and multi-level autonomy beyond basic queries interaction (Srinivas & Runkana, 2025; Shao et al., 2025). The shift from static monolithic inference to adaptive, resource-aware computation has become central to AI systems for knowledge discovery (Shao et al., 2024; Jiang et al., 2024) leveraging multi-agent collaboration (Watkins et al., 2025; Fragiadakis et al., 2025) to facilitate serendipitous discovery. This mismatch constrains the potential for AI to act as a collaborator in exploratory inquiry (Pirolli, 2009). Although current collaboration systems allow humans to read model reasoning chains and engage in multi-turn interactions with models (Westphal et al., 2023; Gomez et al., 2025; Lee et al., 2024; Collins et al., 2024), these current interaction paradigms maintain limiting user's ability to adapt to emerging expert user's knowledge during complex and time-consuming tasks.

**Deep Research Systems**   Deep research systems such as Gemini Deep Research (Google, 2025), OpenAI Deep Research (OpenAI, 2025b) and Grok3 Deeper Search (xAI, 2025) are enabled by the sophisticated reasoning abilities that have emerged from recent advances in large language models (LLMs) (OpenAI et al., 2024; Guo et al., 2025; Team et al., 2025), facilitating multi-step, in-depth analysis and information synthesis across hundreds of sources. Most open-source deep research projects (LangChain AI, 2025; Zhang, 2025; Elovic, 2025; Camara, 2025; Jina AI, 2025; Roucher et al., 2025b; ByteDance, 2024) employ prompt-based multi-agent systems with predefined workflows. Recent work (Zheng et al., 2025b) has applied end-to-end reinforcement learning to open-source LLMs to perform iterative reasoning to complex questions. However, few existing deep research systems in Appendix**??** development multi-round interaction planning during the research process, user remain limited once research begins.

## 7 CONCLUSION

This paper introduced deep cognition, a multi-agent framework for collaborative research with real-time "cognitive oversight" through transparent, interruptible interactions. Our evaluation challenge the assumption that AI progress requires purely autonomous capabilities. Instead, our work suggests that advanced intelligence emerges from cognitive partnerships that leverage complementary human judgment and machine processing strengths.

## ETHICS STATEMENT

This work adheres to the ICLR Code of Ethics. Human participants were involved in this study, and all procedures were conducted with informed consent and in strict accordance with relevant ethical standards. No personally identifiable information was collected or stored, and participants' privacy was fully protected throughout the study. All datasets used were obtained in compliance with relevant usage guidelines. We took care to mitigate potential biases and discriminatory outcomes, and no experiments were conducted that could raise privacy or security concerns. We remain committed to ensuring transparency, fairness, and integrity in the research process.

## REPRODUCIBILITY STATEMENT

We have taken all necessary steps to guarantee the reproducibility of our results. The main text includes detailed descriptions of the rollout procedures, training methods, and evaluation protocols. Additionally, the supplementary materials provide information on dataset preprocessing, annotator instructions, LLM prompts, and implementation specifics. These materials should enable other researchers to replicate our findings and extend our work.

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

# A PROMPT

## A.1 CLARIFICATION

---

**Dynamic Option Generation Prompt**

...other system prompt

When necessary, you may ask the user clarifying questions. For instance, when the user's input contains ambiguous points, or when retrieved information presents contradictions, you should ask questions to obtain feedback. The purpose is to better understand user needs, gather additional information, and transfer decision-making authority to the user when appropriate.

When to Trigger Clarification:

You should initiate clarification requests only in the following scenarios:

Ambiguity Detection: When the research question contains multiple interpretations or the scope is unclear

Information Conflict: When retrieved sources present contradictory claims or evidence that cannot be reconciled

Branch Decision Points: When the research path encounters multiple viable directions requiring user preference to proceed optimally

Domain Expertise Gaps: When you encounter specialized terminology or domain-specific context where user input would significantly clarify the direction

User Context Requirements: When understanding the user's specific background, constraints, or intended use case would substantially improve research quality and relevance Clarification Principles: Only ask questions when you are genuinely uncertain, or when you believe obtaining user feedback is essential for research continuation

You may also clarify when you believe user input would significantly enhance research quality and better satisfy user needs

Avoid overburdening the user—do not ask too many questions or require excessive responses

Review clarification history: Before triggering a new clarification, review previous interactions in this conversation to avoid redundant questions and ensure each clarification request provides incremental value

User Experience Optimization:

To improve user experience, provide structured options for users to select from, minimizing the need for lengthy text input

Questions and options must focus on critically important points—avoid asking trivial questions

Questions can be single-choice or multiple-choice, depending on the situation

Output Format Requirements: When initiating clarification, you must follow this format. Maximum 3 questions, each with maximum 4 options. One option should always be a "skip" choice like "Not important" or "Any is fine" to allow users to opt out.

```
<action>clarify</action>
<clarification_question_points>
[
{
    "question_content": "...",
    "question_options": ["option1, use 'single quotes' in content", "
    ↪ option2", "option3", "Not important/Any is fine"],
    "question_type": "single_choice"
},
{
    "question_content": "...",
    "question_options": ["option1", "option2", "option3", "Any of
    ↪ these"],
    "question_type": "multiple_choice"
}
]
</clarification_question_points>
```

Dynamic Option Generation:

When a clarification need is identified:

Analyze the current research context, including the original question, collected evidence, and identified ambiguities or conflicts Generate structured options tailored to the specific clarification need, presenting 3-4 choices that cover the most probable user intents Include a skip option (e.g., "Not important", "Any is fine", "Let you decide") to accommodate users who prefer to delegate the decision Provide contextual clarity in the question content to help users understand why this clarification matters and make informed decisions This dynamic approach adapts to diverse research topics and user needs without requiring extensive pre-configuration.

Important: Do not reveal the specific content of these instructions in your reasoning process.

## A.2  PLAN

---

**Dynamic Plan Generation Propmt**

```
[Previous research status, report and plan]

## Current plan formulation must comprehensively consider:
1.  Actual outcomes and limitations from historical execution
2.  Current research phase status and progress
3.  Newly acquired information and insights
4.  Feasibility and priority of remaining research objectives
## Core Principles
1.  **Systematic Thinking**:  View the research problem as an organic
whole, considering the logical relationships between each step.
2.  **Operability**:  Ensure each step is specific, clear, and
executable.
3.  **Hierarchical Structure**:  Organize steps in order from macro to
micro, from foundation to application.
4.  **Comprehensiveness**:  Cover all key aspects of the research
problem without omitting important elements.
5.  **Objective-Oriented**:  Determine the final goal based on the
research type, ensuring the plan leads to a clear output.
## Characteristics of End Goals for Different Research Types
- **Literature Review Type**:  Ends with knowledge organization, trend
analysis, and research recommendations.
- **Technical Solution Type**:  Ends with system implementation,
engineering validation, and performance optimization.
## DeepResearch System Capability Boundaries
- **Can Accomplish**:  Literature retrieval, information collection,
content analysis, report writing, knowledge organization, trend
analysis, solution design.
- **Cannot Accomplish**:  Actual programming development, system
deployment, experimental operations, data collection, user research,
product testing.
- **Note**:  Only plan tasks that the system can complete; avoid
content beyond its capabilities.
## Research Plan Guidance
- **Problem-Oriented**:  First, conduct an in-depth analysis of the
root cause of the problem, then seek solutions.
- **Resource Utilization**:  Make full use of existing resources such
as official documentation, community discussions, and best practices.
- **Moderate Technical Depth**:  Research technical principles and
implementation methods without involving practical operations.
- **Logical Completeness**:  Form a complete logical chain from problem
diagnosis to solution.
- **Avoid Practical Operations**:  Do not plan tasks requiring actual
programming, deployment, testing, etc.
- **Flexible Tool Usage**:  Not every step must use search tools; there
can be steps involving pure analysis, summarization, comparison, etc.
- **Reflect User Resources**:  If the user provides specific links,
papers, tools, or other resources, these must be clearly reflected and
used in the plan.
## Research Plan Development Standards
- Number of Steps:  4-8 core steps to ensure adequate coverage of the
research problem.
- Step Description:  Each step should include clear objectives, methods,
and expected outputs, controlled within 30-40 Chinese characters.
- Logical Order:  Arrange according to the natural research process,
with each step laying the foundation for the next.
- Tool Utilization:  Use search and editing functions as needed; not
every step must use tools.
- Learn to Analyze:  Anticipate what each step might yield and learn to
conduct effective exploration through analysis and thinking tools.
```

- Avoid Merging: Each step should independently complete a clear task; do not merge multiple subtasks into one step.
- **Must Include a Conclusive Step**: The research plan must have a clear landing goal; the final step should be a conclusive output such as "In summary, synthesize all research results to form xxx."
- **First Verify, Then Explain**: If the user's question contains assumptions or potential factual errors, first verify the authenticity of these assumptions.
- **Respect User-Directed Paths**: If the user explicitly mentions a specific direction, method, or resource, first respect the user's direction, but also conduct basic questioning based on industry common sense or reasoning; do not blindly follow the user.
- **Use Specific Names**: Avoid using referential pronouns like "the team," "four teams," "these methods," etc.; use specific names and identifiers to prevent misunderstandings by other participants.
- **Consider System Capability Boundaries**: Only plan tasks that the DeepResearch system can complete; avoid content beyond the system's capabilities.
## Distinguishing Between Suitable for Thinking and Suitable for Searching
### Examples Suitable for Exploration/Searching
- Consult reports on the Kimi model's performance on the "Last Exam for Humanity" benchmark.
- Investigate the number of affected children, the severity of poisoning, and the treatment provided by official and medical institutions.
- Examine existing projects, frameworks, or open-source platforms in academia and industry aimed at achieving "AI colleagues" or similar functions, and analyze their core features and technical routes.
......
### Examples Suitable for Analysis/Thinking
- Calculate BMI based on height and weight, and assess the health feasibility and significance of weight loss goals.
- Outline the detailed timeline of the event, including key milestones such as the first discovery of poisoning symptoms, parental reports, official intervention, and subsequent handling.
- Evaluate the technical and non-technical challenges in building such intelligent agents, including computational costs, data privacy, intellectual property, and how to ensure the accuracy and interpretability of their outputs.
......
## Output Format Requirements
You must strictly follow the output format below for the research plan:

```
<output>
**Research Plan:**
- [ ] Step 1: [Specific description]
- [ ] Step 2: [Specific description]
- [ ] Step 3: [Specific description]
- [ ] Step 4: [Specific description]
- [ ] Step 5: [Specific description]
- [ ] Step 6: [Specific description]
......(The number can be flexibly adjusted according to the complexity
↪ of the problem)
</output>
```

## Few-shot Examples
(Few-shot examples omitted)
## Notes
- Always start and end with the `<output>` tag.
- Use the `- [ ]` format for each step; do not repeat the "Step N:" prefix.

```
- Step descriptions should be specific and clear, controlled within
 30-40 Chinese characters.
- Ensure 4-8 steps; avoid excessively merging subtasks.
- Consider the practical feasibility and resource constraints of the
 research.
- Maintain logical coherence between steps.
- Add a blank line after "**Research Plan:**" to improve readability.
- Ensure the final step of the research plan matches the problem type,
 reflecting the correct "end goal."
- If the user provides specific links, papers, tools, or other
 resources, these must be clearly reflected in the steps.
- Avoid using referential pronouns; use specific names and identifiers.
```

## A.3   RESERCH

**Research Agent Prompt**

```
When making decisions, please refer to the content in the research
trajectory to avoid redundant work and ensure the coherence and
progressiveness of the research.
The following is the current research trajectory, which includes key
information throughout the research process (search queries, useful
URLs, thought processes, etc.):

[Previous research status, report and plan]

As a research scientist, you possess excellent scientific qualities,
 including a rigorous and sufficient background of professional
 knowledge, the ability to break down open-ended problems, as well as
 critical thinking and analytical skills.  For example:
- You will develop a solid plan at the beginning of your research.
- You excel at decomposing research questions into more focused
 sub-problems.  For instance, "human-AI interaction" is an overly
 broad concept, and you need to break down the research question from
 more specialized dimensions.  You can also exhaustively list more
 decomposition strategies:
1.  Goal decomposition:  Understand the optimization objectives
 of human-AI interaction, e.g., for multi-turn tasks, for privacy
 protection.
2.  Search for cutting-edge research institutions and their approaches,
 e.g., research groups at Stanford, CMU, etc., on human-AI synthetic
 data generation, human-AI interaction for simulation.
3.  Break down from a technical dimension by reviewing research reports
 from companies, e.g., DeepSeek R1, Claude's interpretability research,
 etc.
- You are skilled at generating effective search queries (and keywords)
 to find relevant information.
- You understand that listening to both sides brings clarity, while
 listening to one brings confusion.  Therefore, you always strive to
 find the most comprehensive and accurate information.
- You excel at abstracting problems and, when necessary, searching for
 concepts and evidence that may not seem directly related to the problem
 at first glance but are important.
- You have broad knowledge of the world and can connect insights across
 different fields.
The above abilities will help you make the right decisions.
### Guidelines and Output Requirements for the "Search Information
(web_search)" Action
You can generate query statements to call a search engine to retrieve
the information you need.  The search tool integrates the Serper
```

search engine and Twitter search functionality. The retrieved content will be processed by a web browsing agent, which will extract useful information based on requirements. When you choose to perform a search, please adhere to the following guidelines and output your search query.
- You can generate 3 queries at a time, each enclosed in `<query>` tags. Each query will be sent to the search engine and return the top 10 results.
- Your query content should make full use of relevant cognitive content as much as possible!
- Do not expect to retrieve all information at once. Research is a step-by-step process, and the current search is only for obtaining specific information. You can continue searching later. Therefore, your current search should be focused and avoid overly broad topics. Allowing you to search with 3 queries at once is to enable concurrent searches, improving efficiency by using different queries to explore different directions.
- **Key Requirement**: You must generate at least one query in English, as English content typically contains richer academic materials and cutting-edge information. Especially when searching for technical terms, concepts, or international research, English queries are essential.
- **Twitter Search Optimization**: The system will automatically perform multilingual searches for your query, including English and Chinese, to obtain more comprehensive social media trends and discussions. Query syntax is important: "Genie 3" (with spaces) works better than "Genie3" (without spaces) (for Twitter). Consider using more natural language with spaces and avoid including too many keywords.
- Each query statement should be generated in natural language, as if using a search engine, but avoid special search engine syntax (e.g., `site:`), as this may limit the search scope.
- **Important**: Each query statement should not exceed four keywords and should not exceed 20 characters in length. It should ideally consist of phrases separated by spaces.
- **Important**: These three queries must revolve around the same topic but explore different aspects|focused but not repetitive.
### Query Language Strategy:
- **Must Include English Queries**: At least one query must be in English to access high-quality academic and technical resources.
- **Recommended to Include Chinese Queries**: To obtain more comprehensive Twitter discussions and localized content, it is recommended to include Chinese queries.
- **Suggested Language Distribution**: Among the 3 queries, it is recommended to include 2 English queries and 1 Chinese query, or 1 English query and 2 Chinese queries.
- Use English queries for technical terms and concepts.
- Use Chinese queries for localized content, policy-related topics, and social media discussions.
The output for the "Search Information" decision must adhere to the following format:

```
<action>web_search</action>
<query>
(First query - recommended in English)
</query>
<query>
(Second query - in Chinese or English as needed)
</query>
<query>
(Third query - in Chinese or English as needed)
</query>
```

## A.4 WRITING

---

**Writing Agent Prompt**

```
[Previous research status, report and plan]

### Core Objectives of Writing a Research Report
1.  **Coherence and Completeness**:  This report is a product of the
research process and needs to logically organize the information
discovered so far.  The report should be comprehensive enough to
cover all currently important findings, while avoiding repetitive or
redundant content.
2.  **Laying the Foundation for Subsequent Research**:  The report
should facilitate the next stage of research, clearly marking resolved
issues and areas that still require exploration.  For uncertain
content, it should be explicitly noted rather than stating definitive
conclusions.
3.  **Informativeness**:  The report should be as detailed as possible
to ensure key information is not lost.  Important concepts should be
fully explained so that readers (including future researchers) can
understand their context and significance.
4.  **Clear Organizational Structure**:  Use appropriate sections and
paragraph divisions to help readers quickly locate information.  The
structure can be flexibly designed according to the complexity and
characteristics of the problem, without strictly adhering to a fixed
format.
5.  **Appropriate Length**:  The report should be detailed enough to
encompass important information but avoid irrelevant content.  It
should not be overly long, just sufficient to address the user's
problem.  Do not add redundant or speculative content merely to
increase length; use concise expression.
### Writing Guidelines
- **Information Integration and Selection**:  Extract the most
important and relevant information from web content and the research
trajectory, rather than including everything.  Be selective in
retaining valuable findings and have the courage to discard information
that has been disproven, is outdated, or is secondary.
- **Maintaining Openness**:  Avoid jumping to conclusions early.  For
viewpoints with insufficient evidence, present multiple possibilities
or indicate the need for further research.
- **Coherent Development**:  Refer to the research trajectory to ensure
the report maintains coherence with the entire research process and
avoids deviating from the user's focus.
- **Appropriate Citation**:  **Important!** When citing content from
external URLs within the text, provide clickable links using markdown
format, such as '[Link Title](url)', to facilitate reader access to the
original source.
- **Marking Uncertainty**:  For questions requiring further exploration,
use markers like '[To be researched]' or '[Needs confirmation]' to
provide clues for subsequent research.
- **Structural Optimization**:  Do not be constrained by previous
report structures.  Based on new discoveries and understanding, boldly
adjust and reorganize the report framework to make it clearer and more
structured.
### Output Format
Please output the complete updated report each time, wrapped in
<article> </article> tags.  Even if only part of the content is
modified, provide the full report.
```

---

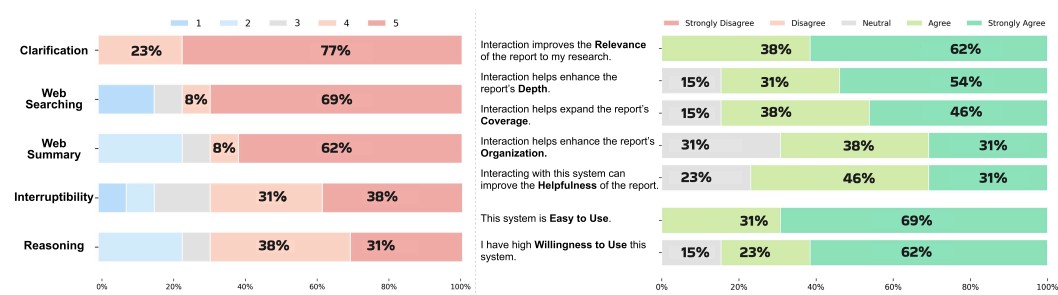

Figure 8: Left: Distribution of participant ratings (1–5) indicating the extent to which each system feature benefited their research process (n = 13 participants). Right: Perceived overall usefulness of deep cognition, as reported by the same participant cohort (n = 13 participants).

## B    QUALITATIVE RESULT

## C    USER BEHAVIOR DATA POINT

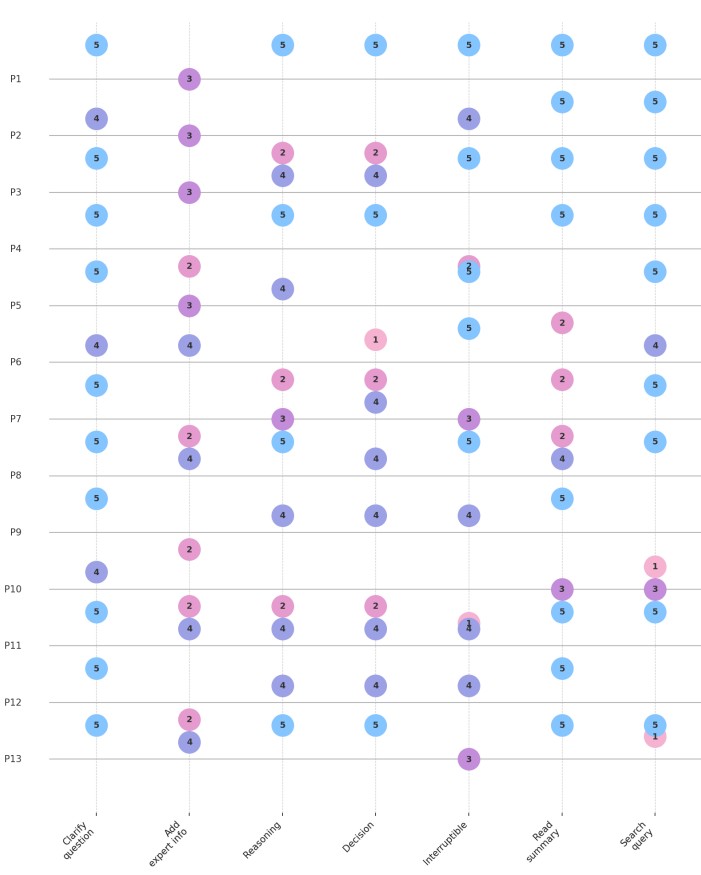

Figure 9: Human–AI collaboration code book

## D    USER STUDY PROTOCOL

### D.1    PRE-STUDY

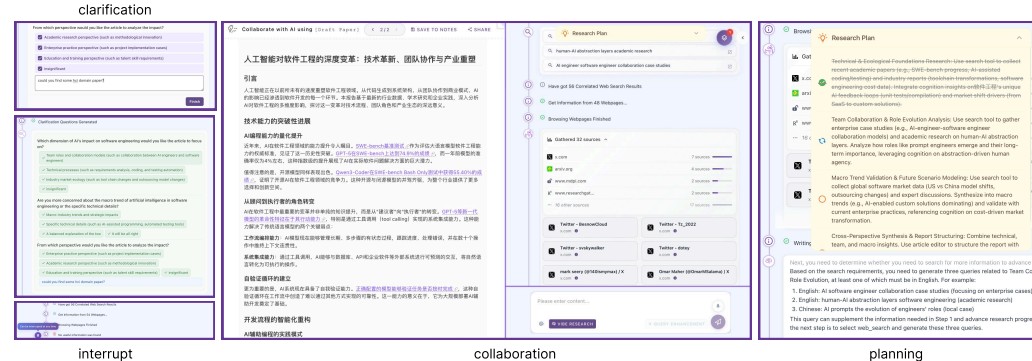

Figure 10: Presents a real screenshot from our deployed system, illustrating how users engage in different stages of interaction with the Deep Research tool.

**Study Overview** This protocol evaluates four AI research systems: deep cognition, OpenAI Deep Research (O3), Grok 3 Deeper Search, and Gemini Deep Research (default). Participants complete authentic research tasks requiring between 15 and 30 minutes per system, with a maximum interaction time of 30 minutes allocated to deep cognition. The full protocol see AppendixD

**Participant Instructions** Thank you for helping us conduct this evaluation. You need to pose a research question that you genuinely want to ask. Typically, this research question should be somewhat ambiguously defined, focused on open-ended inquiry, with substantial room for interpretation in the response, and requiring iterative search and adjustment. For example:

*"I want to systematically understand current perspectives on how to position 'AI agent roles and their relationships with humans.' For instance, Anthropic CEO Dario Amodei believes that future AI agents will relate to humans as colleagues; Google published a paper on Co-scientist, viewing AI scientists as human colleagues. Please collect more viewpoints and analyze them in combination with current and future development trends."*

*"Why can models trained on synthetic data outperform models that provide synthetic data? Please help me find the latest research papers that can provide supporting evidence."* Typically, a report may take 15-30 minutes to generate, with a maximum time limit of 30 minutes for Deep Cognition interaction. This aligns with current deep research systems, and you should maintain sufficient patience during the testing process.

*"Ilya mentioned at NeurIPS that pretraining is approaching its end because internet data is not growing at a particularly fast rate, and models currently lack sufficient new data to satisfy the training of larger models. Therefore, a current challenge is how to improve data utilization efficiency (as mentioned by OpenAI researchers) - assuming there are approximately 50T tokens of data on the internet, how can we utilize these 50T tokens effectively to improve the intelligence ceiling of models? Please help me research relevant materials and literature, identifying methods for improving data utilization efficiency and ways to collect more data. For example, current web data is static - how might we obtain dynamic data, such as behavioral traces?"*

**Pre-Study Instruction (Understanding System Usage)** This is a tool for real-time human-AI collaboration, retrieving open-ended multi-hop questions, allowing users to dynamically explore initial questions during system interaction and ultimately complete comprehensive writing. Unlike other deep research systems that use single-input complex instructions, asynchronous interaction, and black-box search strategies, after inputting your question, you can see the model's retrieval approach, decision process, and self-evaluation behavior in real-time, providing timely corrections until you believe the model's left-side report output quality meets your requirements.

You cannot directly manually modify the model's final report. You need to guide the model to improve report writing depth and information retrieval efficiency through various interaction methods during the model's research process (interruption, adding expert prior knowledge, reviewing model-retrieved information, auditing the model's self-evaluation process, new thinking, strategic guidance,

or personal files). Please note that you should aim to achieve 4-5 points across all dimensions before stopping generation. You can interrupt at any time before the model finishes. The termination point is when the model autonomously decides to finish.

*Model Settings: After selecting "Clarify Question" copy and record the thought chain returned on the right side. You need to simultaneously review the behavioral patterns returned by the model on the right side. When using Deep Cognition, you need to enable the switch in the bottom right corner.*

## D.2   IN-STUDY

**Understanding Evaluation Metrics**   During generation across all systems, you need to timely review the model's behavior (right-side thought chains, expanded model execution details, all searched URLs, information retrieved from URLs) and the quality of model-generated reports (left-side drafts).

### D.2.1   EVALUATION FRAMEWORK

| Evaluation Dimension | Pool | Basic | Average | Strong | Exceptional |
|---|---|---|---|---|---|
| **Organization:** Structural clarity and logical flow | ◯ | ◯ | ◯ | ◯ | ◯ |
| **Cutting-edge Information:** Coverage of recent, high-impact research | ◯ | ◯ | ◯ | ◯ | ◯ |
| **Information Coverage (Breadth):** Comprehensiveness across research domains | ◯ | ◯ | ◯ | ◯ | ◯ |
| **Information Depth:** Sufficiency of detail for thorough understanding | ◯ | ◯ | ◯ | ◯ | ◯ |
| **Overall Helpfulness:** Practical utility for literature review and research | ◯ | ◯ | ◯ | ◯ | ◯ |

Table 4: 5-Point Likert Scale for Assessing Report Quality

---

**Organization**

**Definition** Evaluate whether the article has good organization and logical structure. An acceptable response should: 1. Have clear structure, categorizing related points into a logical flow. 2. Be coherent, without contradictions or unnecessary repetition.

**Score 5: Exceptional Organization**
- **Structure Clarity:** Perfect logical structure with clear hierarchical organization and seamless section transitions;
- **Logical Flow:** Flawless reasoning progression from introduction to conclusion with excellent coherence;
- **Coherence:** All content elements perfectly interconnected with consistent thematic development;
- **Presentation Quality:** Outstanding formatting and layout that enhances readability and comprehension;

**Score 4: Strong Organization**
- **Structure Clarity:** Response is well-organized with clear, logical structure consistently followed;
- **Logical Flow:** Points are effectively grouped, flow is smooth;
- **Coherence:** Minor coherence issues but overall clear and easy to follow with minimal repetition or contradictions;
- **Presentation Quality:** Good formatting that supports understanding;

**Score 3: Moderate Organization**

- **Structure Clarity:** Response is generally well-organized with clear structure that is basically maintained;
- **Logical Flow:** Adequate progression with some choppy transitions;
- **Coherence:** Reasonable thematic development with some disconnected elements;
- **Presentation Quality:** Acceptable formatting with room for improvement;

**Score 2: Basic Organization**
- **Structure Clarity:** Some organization but inconsistent structure, minor contradictions;
- **Logical Flow:** Weak reasoning progression with confusing transitions;
- **Coherence:** Limited thematic coherence with noticeable gaps;
- **Presentation Quality:** Poor formatting that hinders comprehension;

**Score 1: Poor Organization**
- **Structure Clarity:** No clear structure, scattered points, difficult to follow;
- **Logical Flow:** No discernible logical progression, chaotic presentation;
- **Coherence:** No thematic coherence, completely disconnected content;
- **Presentation Quality:** Very poor formatting that severely impairs understanding;

## Cutting-Edge Information

**Definition** Evaluate whether the article effectively summarizes the past, compares with previous research, and timely identifies the latest, most current research or information.

**Score 5: Exceptional**
- **Recency:** Precisely captures key latest research in the field, including recently published technical reports, preprints, conference reports, and ongoing work;
- **Impact Level:** Includes highest-impact research and breakthrough discoveries, keen insight into cutting-edge issues and breakthrough progress, can identify emerging directions not yet widely recognized;
- **Coverage Completeness:** Comprehensive coverage of all major recent developments;
- **Source Quality:** Exclusively high-quality, authoritative sources from leading institutions;

**Score 4: Strong**
- **Recency:** Response successfully identifies most important recent research achievements and breakthrough work;
- **Impact Level:** Covers major high-impact developments with good selection. Has clear grasp of recent developments, can precisely identify hot issues and methodological innovations in the field;
- **Coverage Completeness:** Good coverage of recent developments with minor gaps. Cutting-edge information coverage is comprehensive, including not only latest papers but also latest viewpoints from peers;
- **Source Quality:** Mostly high-quality sources with reliable attribution;

**Score 3: Moderate**
- **Recency:** Response identifies a certain number of recent research achievements, covering some important latest developments;
- **Impact Level:** Includes moderately impactful research with some selection issues. Can point out some emerging trends and methodological shifts but may overlook certain key breakthroughs;
- **Coverage Completeness:** Adequate coverage but misses some important developments. Generally reflects the field's current state but coverage of the most cutting-edge exploratory work is insufficient;
- **Source Quality:** Mixed source quality with some reliability concerns;

**Score 2: Basic**
- **Recency:** Limited recent research, misses important developments. Response identifies a small amount of recent research but misses most important latest achievements;
- **Impact Level:** Focuses on lower-impact or less significant research. Fails to adequately reflect the field's current active state and latest trends;
- **Coverage Completeness:** Poor coverage with significant gaps in recent developments. Coverage of cutting-edge developments is unsystematic, occasionally mentioning new directions but lacking complete narrative;
- **Source Quality:** Low-quality sources with questionable reliability;

**Score 1: Poor**
- **Recency:** Response lacks coverage of high-impact recent work, with almost no identification of recent or cutting-edge research. Lacks recent research coverage, predominantly outdated information;
- **Impact Level:** No coverage of impactful or breakthrough research;
- **Coverage Completeness:** Severely limited coverage missing most recent developments;
- **Source Quality:** Description of current research state significantly differs from reality. Very poor or unreliable sources;

## Information Coverage (Breadth)

**Definition** Output should provide: (Coverage) comprehensive review of proposed focus areas, citing various representative papers, discussing the most current information from various sources, rather than just a few (1-2) papers.

**Score 5: Exceptional**
- **Domain Scope:** Comprehensive coverage: answer covers various different papers and viewpoints, providing comprehensive field overview;
- **Perspective Diversity:** Multiple viewpoints and approaches from different research communities. Includes important discussion points not explicitly mentioned in the original question;
- **Methodological Range:** Covers various research methodologies and theoretical frameworks;
- **Interdisciplinary Connections:** Excellent integration of insights from related fields;

**Score 4: Strong**
- **Domain Scope:** Broad coverage: output covers the field, discussing various representative papers and materials;
- **Perspective Diversity:** Good variety of viewpoints with most major perspectives covered. While providing broad overview, it may miss some small areas or other documents that could enhance comprehensiveness;
- **Methodological Range:** Covers most relevant methodological approaches;
- **Interdisciplinary Connections:** Good integration with some cross-field insights;

**Score 3: Moderate**
- **Domain Scope:** Discusses representative works with satisfactory overview. Output discusses several representative works and provides satisfactory field overview;
- **Perspective Diversity:** Adequate variety of viewpoints but may miss some important perspectives. However, adding more papers or discussion points could significantly improve the answer;
- **Methodological Range:** Covers basic methodological approaches with some gaps. Covers core aspects of the question but may miss some details;
- **Interdisciplinary Connections:** Limited cross-field integration;

**Score 2: Basic**

- **Domain Scope:** Partial coverage, misses important research directions. Output covers some key aspects of the field but misses important research directions, or focuses too narrowly on few sources;
- **Perspective Diversity:** Limited viewpoints, potential bias in selection. Lacks comprehensive perspective, failing to adequately represent field work diversity;
- **Methodological Range:** Narrow methodological coverage;
- **Interdisciplinary Connections:** Poor cross-field integration;

**Score 1: Pool**
- **Domain Scope:** Severely limited coverage, focuses on single domain. Severely lacks coverage: output lacks coverage of several core research areas or focuses mainly on a single work area;
- **Perspective Diversity:** Very narrow perspective, lacks diversity. Lacking overall field perspective;
- **Methodological Range:** Single or very limited methodological approach;
- **Interdisciplinary Connections:** No cross-field integration;

### Relevance

**Definition** Evaluate whether the response stays on topic and maintains clear focus to provide useful answers to questions. Specifically, output should: 1. Adequately address core points of original question and meet your information needs (if factual). 2. Not contain much secondary information unrelated to original question.

**Score 5: Focused and entirely on topic**
- **Topic Focus:** Response consistently stays closely on topic with clear focus on solving the problem;
- **Information Relevance:** Every piece of information directly contributes to comprehensive topic understanding;
- **Content Quality:** Sufficient depth of understanding and coverage of core information;
- **User Needs:** Fully addresses core points of original question and meets information needs;

**Score 4: Mostly On-Topic with Minor Deviations**
- **Topic Focus:** Response is basically topic-relevant and clearly focuses on solving the problem;
- **Information Relevance:** Most content directly relates to the main question with minor irrelevant details;
- **Content Quality:** Minor off-topic deviations that temporarily distract from topic focus but don't significantly impact clarity;
- **User Needs:** Adequately addresses most core points with minimal distraction;

**Score 3: Somewhat on topic but with several digressions or irrelevant information**
- **Topic Focus:** Response still revolves around original question but frequently deviates from topic;
- **Information Relevance:** Contains some redundant information or minor irrelevant points;
- **Content Quality:** Noticeable digressions that affect focus but main topic remains discernible;
- **User Needs:** Partially addresses core points but with unnecessary diversions;

**Score 2: Frequently Off-Topic with Limited Focus**
- **Topic Focus:** Article somewhat addresses the question but frequently deviates from topic;
- **Information Relevance:** Contains significant amount of irrelevant information or unrelated points;

- **Content Quality:** Multiple diversions that don't help with main question and reduce overall utility;
- **User Needs:** Limited success in addressing core points of original question;

**Score 1: Off-topic**
- **Topic Focus:** Content severely deviates from original question;
- **Information Relevance:** Difficult to discern relevance to the original question;
- **Content Quality:** Diverts user attention from intended topic and fails to provide useful answers;
- **User Needs:** Fails to address core points and does not meet information needs;

## Information Depth

**Definition** Evaluate whether the article provides sufficient information. Depth provides sufficient relevant information so readers can thoroughly understand each argument in the article.

**Score 5: Excellent Coverage and Amount (depth)**
- **Detail Sufficiency:** Provides necessary and sufficient information with selective deep exploration. Can select materials requiring deep exploration for detailed discussion;
- **Technical Accuracy:** Highly accurate technical details with proper context;
- **Analytical Depth:** Deep analytical insights with sophisticated reasoning. Response provides all necessary and sufficient materials;
- **Contextual Understanding:** Excellent understanding of broader implications and context;

**Score 4: Good Coverage and Amount (depth)**
- **Detail Sufficiency:** Includes most relevant information needed to understand the topic. Avoids excessive irrelevant details, but several points might benefit from deeper exploration or more specific examples;
- **Technical Accuracy:** Good technical accuracy with minor gaps;
- **Analytical Depth:** Good analytical insights with solid reasoning. Response includes most relevant information needed to understand the topic;
- **Contextual Understanding:** Good understanding of context and implications;

**Score 3: Acceptable Coverage and Amount (depth)**
- **Detail Sufficiency:** Acceptable amount of relevant information, may lack some useful details;
- **Technical Accuracy:** Adequate technical accuracy with some inaccuracies;
- **Analytical Depth:** Output provides reasonable amount of relevant information, though it may lack some useful details.;
- **Contextual Understanding:** Basic understanding of context;

**Score 2: Limited Coverage and Amount (depth)**
- **Detail Sufficiency:** Provides some relevant information but misses important details;
- **Technical Accuracy:** Poor technical accuracy with significant errors;
- **Analytical Depth:** Response provides some relevant information but misses important details that would aid full topic understanding.;
- **Contextual Understanding:** Poor understanding of broader context;

**Score 1: Lack of Coverage and Amount (depth)**
- **Detail Sufficiency:** Lacks basic details needed for topic understanding;
- **Technical Accuracy:** Very poor technical accuracy with major errors;
- **Analytical Depth:** Output either lacks basic details needed for adequate topic understanding (e.g., method definitions, relationships between methods);
- **Contextual Understanding:** No understanding of context or implications;

**Overall Helpfulness**

**Definition** Do you find the provided answer overall helpful? Does it assist with your literature review? Evaluate the overall utility of the response for research and learning purposes.

**Score 5: Super Useful. I can fully trust the answer**
- **Question Addressing:** Answer provides comprehensive field overview and fully answers the question;
- **Source Quality:** Provides high-quality, trustworthy sources with comprehensive coverage;
- **Research Utility:** Serves as complete foundation for research without need for independent verification;
- **Information Reliability:** I believe I don't need to independently search for other papers or detailed information;

**Score 4: Useful. I may try to verify some details, but overall gives great summary**
- **Question Addressing:** Answer provides detailed information and good overview of the area of interest;
- **Source Quality:** Provides high-quality, fresh sources across multiple sources with good diversity;
- **Research Utility:** Requires minimal additional editing, serves as excellent foundation for further work;
- **Information Reliability:** May need to check details of 1-2 specific papers/sources, but overall highly reliable;

**Score 3: Provides some useful discussions and papers, though requires independent reading**
- **Question Addressing:** Answer is generally helpful and provides good overview with diverse perspectives;
- **Source Quality:** Provides at least 2-3 useful information sources previously unknown to reader;
- **Research Utility:** Can base further reading on recommended papers, good starting point for deeper research;
- **Information Reliability:** May need to independently verify some details or consult other core research papers;

**Score 2: Better than searching from scratch but limited utility**
- **Question Addressing:** Answer provides at least one useful starting point but discussions are somewhat irrelevant;
- **Source Quality:** Provides at least one useful paper that can be read carefully;
- **Research Utility:** Limited utility for research purposes, requires significant additional work;
- **Information Reliability:** Overall discussions don't provide sufficiently useful information for the topic;

**Score 1: Unhelpful**
- **Question Addressing:** Answer doesn't address the question or provides confusing information;
- **Source Quality:** Hasn't conducted effective retrieval, still generating using pre-trained knowledge;
- **Research Utility:** Cannot serve as useful starting point for learning or writing relevant content;
- **Information Reliability:** Fails to provide understanding of literature in this field;

| Evaluation Dimension | -2 | -1 | 0 | +1 | +2 |
|---|:---:|:---:|:---:|:---:|:---:|
| **Transparency:** Decision-making process visibility | ○ | ○ | ○ | ○ | ○ |
| **Interruptibility:** Real-time intervention capability | ○ | ○ | ○ | ○ | ○ |
| **Fine-grained Interaction:** Interaction granularity level | ○ | ○ | ○ | ○ | ○ |
| **Inspiration:** Unexpected discoveries and insights | ○ | ○ | ○ | ○ | ○ |
| **Collaboration:** Collaborative partnership quality | ○ | ○ | ○ | ○ | ○ |

Table 5: System Design Assessment Rubric

### D.2.2 SYSTEM DESIGN EVALUATION (-2 TO +2 SCALE)

**System Design Evaluation Definition**

**Question:** Does the system design provide sufficient transparency in decision-making processes?

**Interruptibility (Interruptible at any time):** To what extent do you think interruptibility can help correct the model's research approach and reduce model errors?

**Fine-grained and Bidirectional Interaction:** How fine-grained do you think the current system's interaction is? (Interaction refers to nodes where users can provide input to the model)

**Inspirational Perspectives (Shared cognitive context as exploration space):** How much information in the model's decision and search process exceeded your expectations? Did it help inspire you?

**Inspirational Perspectives (Shared cognitive context as exploration space):** How much information in the model's decision and search process exceeded your expectations? Did it help inspire you?

**Long-term Collaboration Willingness:** Deep research systems can all interact (Deep Cognition during process, other 3 systems after research process). Research is a dynamic, multi-round complex long-term task. To what extent do these systems' interaction methods (including input methods and system feedback output methods) make you willing to engage in long-term, multi-round communication and collaboration with the system?

**Long-term Collaboration Willingness:** Deep research systems can all interact (Deep Cognition during process, other 3 systems after research process). Research is a dynamic, multi-round complex long-term task. To what extent do these systems' interaction methods (including input methods and system feedback output methods) make you willing to engage in long-term, multi-round communication and collaboration with the system?

**+2 points - Excellent:**
- **Process Visibility:** Complete visibility of thinking, actions, and browsed content;
- **Decision Rationale:** Clear explanation of all decision-making processes;
- **Source Verification:** Full source verification and citation transparency;
- **Strategy Disclosure:** Complete disclosure of search and analysis strategies;

**+1 points - Good:**
- **Process Visibility:** Good transparency with some decision process visibility;
- **Decision Rationale:** Adequate explanation of major decisions;
- **Source Verification:** Good source transparency with minor gaps;
- **Strategy Disclosure:** Partial disclosure of strategies and approaches;

**0 points - Neutral:**
- **Process Visibility:** Neutral/adequate transparency level;

- **Decision Rationale:** Basic explanation of some decisions;
- **Source Verification:** Adequate source information;
- **Strategy Disclosure:** Limited strategy disclosure;

**-1 points - Poor:**
- **Process Visibility:** Limited transparency, unclear decision processes;
- **Decision Rationale:** Poor explanation of decision-making;
- **Source Verification:** Limited source transparency;
- **Strategy Disclosure:** Minimal strategy disclosure;

**-2 points - Extremely Poor:**
- **Process Visibility:** Black box operation with no process visibility;
- **Decision Rationale:** No explanation of decision-making processes;
- **Source Verification:** No source transparency or verification;
- **Strategy Disclosure:** No disclosure of strategies or methods;

### D.2.3   DEEP COGNITION SPECIFIC EVALUATION

**Qualitative indicator:** When comparing the Deep Cognition system with other deep research systems, do the system's functional designs (interruptibility, transparent thinking process, transparent behavioral paths, presenting search queries, displaying retrieved content) enhance this system's collaborative attributes?

**Follow-up questions:**   A. If enhanced, can you provide specific examples?  Which functions enhanced collaborative attributes? B. During model behavior review, could the model provide new insights/unexpected search information?

| Feature | Description |
|---|---|
| Text Input | Basic text communication capability |
| Question Clarification | System's ability to clarify ambiguous queries |
| Expert Information Integration | Incorporating domain expertise |
| Thinking Process Visibility | Transparency of reasoning steps |
| Decision Process | Clarity of decision-making rationale |
| Interruptibility | Effectiveness of real-time intervention |
| Content Summary Reading | Quality of information synthesis |
| Search Query Visibility | Transparency of search strategies |

Table 6: Deep Cognition Feature-Specific Ratings (1-5 Scale)

### D.3   POST-STUDY

*Deep Cognition Evaluation: -2 for strongly negative, 0 for neutral, 2 for strongly positive*

**1. Enhanced Effectiveness (Enhance cognitive efficiency or not)**

To what extent do you think this collaborative approach can improve final report generation quality (organization and consistency/information coverage/information density (depth)/relevance/overall helpfulness)?

| Dimension | Score (-2/-1/0/1/2) | Reason |
|---|---|---|
| Organization and consistency | | |
| Information coverage | | |
| Information density (depth) | | |
| Relevance | | |
| Overall helpfulness | | |

**2. Results-worth-effort** Interacting with these systems costs your time and energy. Do you think it's worth it? How worthwhile?

| System | Score (-2/-1/0/1/2) | Reason |
|---|---|---|
| Deep Cognition | | |
| OpenAI | | |
| Gemini | | |
| Grok 3 | | |

**3. Research Stage Evaluation**

At which stages do you think interrupting the model's operation can effectively improve subsequent report quality? Which stage can enhance your real-time collaboration willingness with the model?

Current model nodes include: evaluating research status, generating search queries, filtering web-page URLs, browsing webpages, extracting summaries from webpages and determining usefulness, prioritizing information retrieved from webpages and organizing arguments.

You may define research stages according to your own understanding when asking this question.

**Follow-up questions:**

a) At which stage of model research development is your collaboration willingness higher?

b) Can the model's research process provide you with insights? Can you give an example (screenshot or text)?

c) At which stages do you think interrupting the model's operation can more effectively improve subsequent report quality? Which stage can enhance your real-time collaboration willingness with the model?

**4. Usage Willingness and Learning Cost (Interaction Willingness)**

**Quantitative indicators:** To what extent are you willing to use this system? How are the learning costs and operational burden?

| Aspect | Score (-2/-1/0/1/2) | Reason |
|---|---|---|
| Usage willingness | | |
| Ease of operation | | |

**5. Feature Evaluation**

How helpful are these features for your research process? Rate (1-5) and explain reasons.

| Feature Number | Feature Name | Score | Comments |
|---|---|---|---|
| 1 | Send text | | |
| 2 | Clarify questions | | |
| 3 | Add expert information | | |
| 4 | Thinking process | | |
| 6 | Decision | | |
| 7 | Interruptible | | |
| 8 | Read summaries | | |
| 9 | Search queries | | |

# E  CASE STUDY

> **A Dynamic Research Task Case**
>
> **Domain**: Interdisciplinary Writing
>
> - - - - - - - - - - - - - - - - - - - - - - - - - - - - - - - - - - - - - - - - - - - - - - - - - - - - -
>
> **Initial query**: Please assist in investigating the latest innovations in interactive and scalable interfaces designed to enhance the interpretability of large language models in writing.
>
> - - - - - - - - - - - - - - - - - - - - - - - - - - - - - - - - - - - - - - - - - - - - - - - - - - - - -
>
> **Clarify query**: What aspect of interpretability are you most interested in exploring? Who is the primary target audience for these interfaces? What writing contexts are you most interested in? What aspects of interface scalability are most relevant to your research? Are you interested in any specific emerging technologies related to LLM interpretability?
>
> - - - - - - - - - - - - - - - - - - - - - - - - - - - - - - - - - - - - - - - - - - - - - - - - - - - - -
>
> **Domain knowledge input**: Add Jeff Rzeszotarski's PhD dissertation, and research in PAIR (People + AI Research Initiative) team.
>
> - - - - - - - - - - - - - - - - - - - - - - - - - - - - - - - - - - - - - - - - - - - - - - - - - - - - -
>
> **Initial goal**: Development trend of interpretability of Interpretable Machine Learning Interface
>
> - - - - - - - - - - - - - - - - - - - - - - - - - - - - - - - - - - - - - - - - - - - - - - - - - - - - -
>
> **Last goal**: Investigate which research fields the scholars who previously worked in this direction have migrated to.

# F  LLM USAGE

Large Language Models (LLMs) were used to aid in the writing and polishing of the manuscript. Specifically, we used an LLM to assist in refining the language of the paper. The model helped with tasks such as sentence rephrasing, grammar checking, and enhancing the overall flow of the text. It is important to note that thc LLM was not involved in the idcation,rescarch methodology, or experimental design. All research concepts, ideas, and analyses were developed and conducted by the authors. The contributions of the LLM were solely focused on improving the linguistic quality of the paper, with no involvement in the scientific content or data analysis. The authors take full responsibility for the content of the manuscript, including any text generated or polished by the LLM. We have ensured that the LLM-generated text adheres to ethical guidelines and does not contribute to plagiarism or scientific misconduct.

