# OpenReview forum: "Deep Cognition: A Multi-Agent Framework for Collaborative Research with Real-Time Cognitive Oversight"
_ICLR.cc/2026/Conference — ICLR 2026 Conference Withdrawn Submission_

### Official Review · Reviewer_822V · 2025-10-29

**Soundness:** 4
**Presentation:** 3
**Contribution:** 3
**Rating:** 4
**Confidence:** 5

**Summary:**

This paper suggests an extended DeepResearch system where the agent can proactively ask for clarifing questions.

**Strengths:**

It is a nice idea to add proactive dialogue for a system like DeepResearch, breaking the input/think/output paradigm. The results are convincing, and resulting system makes much more sense. This idea is also extensible to other tasks, from Math to Q&A.

**Weaknesses:**

1. There are typos in the paper (one in the section heading), which is embarrassing
2. The abstract says the core is the StateManager architecture and it is never mentioned again the paper again.
3. The authors are totally unaware of dialogue state tracking (DST) literature in the Conversational AI community. They can also simply web search DSTC for building stateful dialogues. They can also read some survey papers on conversational agents, especially on task completion agents.
4. The interaction is reduced to agent asking clarifying questions. This is rather limited and the dialogue can be much richer going beyond clarifying questions. Search for dialogue act tagging to see other categories.

**Questions:**

1. Please also add the prompt for triggering clarifying questions in the Appendix.

---

> ### Author Response · Authors · 2025-11-27
>
> We sincerely thank the reviewer for their careful reading and insightful evaluation of our work. The key issues raised are highly valuable in helping us improve both the overall quality and the technical clarity of the paper.
>
> **Comment 1**: There are typos in the paper (one in the section heading), which is embarrassing.
>
> **Response to comment 1**: We fix it.
>
> **Comment 2**: The abstract says the core is the StateManager architecture and it is never mentioned again the paper again.
>
> **Response to comment 2**: We rewrite our abstract and core framework description and fix it.
>
> **Comment 3**: The authors are totally unaware of dialogue state tracking (DST) literature in the Conversational AI community. They can also simply web search DSTC for building stateful dialogues. They can also read some survey papers on conversational agents, especially on task completion agents.
>
> **Response to comment 3**: Thank you for highlighting the relevant literature on Dialogue State Tracking (DST) and Conversational AI. We agree that this field provides a valuable foundation for managing stateful dialogue. We plan to incorporate this discussion into the **Related Work** section. Our primary focus in this paper, however, is on introducing a transparent multi-agent framework that enables fine-grain human interaction.
>
> **Comment 4**: The interaction is reduced to agent asking clarifying questions. This is rather limited and the dialogue can be much richer going beyond clarifying questions. Search for dialogue act tagging to see other categories.
>
> **Response to comment 4**: We appreciate the suggestion regarding Dialogue Act Tagging and the potential for richer interaction. We want to clarify that our interaction model is not limited to clarification questions alone. The paper introduces a comprehensive framework that supports **several detailed interaction types beyond simple clarification**, including: **Real-Time Intervention (pausing/redirecting the live research at any round), Fine-Grained Interaction (Intermediate research reports through a writing agent，editing the intermediate draft, click and selecting specific URLs), User Knowledge Input (injecting prior expertise/documents), and Transparent Reasoning (user can review of search queries and agent model decision-making processes)**. We have **revised the methodology section to emphasize the completeness of these interactive features**.
>
> **Comment 5**: Please also add the prompt for triggering clarifying questions in the Appendix.
>
> **Response to Comment 5**: We have added **the prompt for triggering clarifying questions**, and **all other core system prompts,** to Appendix.

---

### Official Review · Reviewer_uzS1 · 2025-10-31

**Soundness:** 2
**Presentation:** 3
**Contribution:** 2
**Rating:** 2
**Confidence:** 3

**Summary:**

This paper presents Deep Cognition, a multi-agent system for collaborative deep research that enables real-time human intervention during AI-driven information retrieval and synthesis.

**Strengths:**

The paper attempts to address user control in automated research systems, which is a reasonable area of investigation. The evaluation framework covers multiple dimensions of both output quality and user experience, demonstrating effort in comprehensive assessment.

**Weaknesses:**

### Fundamental Mischaracterization of Baseline Systems

The paper's central premise that existing deep research systems lack transparency, interruptibility, and real-time interaction capabilities is factually incorrect. Current commercial systems already provide the core features that Deep Cognition claims as innovations:

OpenAI Deep Research and ChatGPT provide a stop/pause button that allows users to halt generation at any point. ChatGPT's interface shows streaming output in real-time, enabling users to interrupt when they observe undesirable directions. After interruption, users can provide corrective feedback and request continuation or modification. Claude (Anthropic) includes thinking tags that expose intermediate reasoning steps. Claude Sonnet's extended thinking mode shows the model's internal deliberation process. Users can review these reasoning traces and provide feedback to redirect the conversation. Perplexity AI displays search queries as they are generated, shows which sources are being accessed, and provides real-time transparency into the retrieval process. Users can see the information gathering happening and can follow up with refinements. Gemini similarly shows progressive output generation and allows users to intervene with additional instructions or corrections during multi-step tasks.

Given these capabilities in existing systems, the paper's claimed innovations reduce to minor interface variations rather than fundamental capabilities. The "cognitive oversight" paradigm is not a new interaction model but rather a repackaging of features already present in commercial products. This fundamentally undermines the paper's contribution claims.

### Invalid Experimental Comparisons

The experimental design compares Deep Cognition's interactive mode against baseline systems used in constrained, non-interactive ways. This comparison is invalid for multiple reasons:

First, the baselines are evaluated through their API interfaces or limited testing protocols that deliberately exclude their interactive capabilities. The paper does not test OpenAI Deep Research or ChatGPT used as intended—with users providing initial queries, reviewing outputs, interrupting when necessary, and engaging in multi-turn refinement conversations. Instead, baselines are apparently restricted to single-input scenarios, creating an artificial disadvantage.

Second, the comparison confounds interaction capability with interaction amount. Deep Cognition involves continuous user monitoring for up to 30 minutes with multiple intervention points. Baseline systems are tested without equivalent interaction opportunities. Any performance difference could simply reflect more total user input rather than system design. A fair comparison would equalize either (1) total interaction time, (2) number of feedback rounds, or (3) amount of user guidance provided across all conditions.

Third, the paper claims baselines use an "input-wait-output" paradigm as a fundamental limitation. This misrepresents product design choices as technical constraints. These systems adopt streamlined interaction models because many users prefer convenience over control. The paper provides no evidence that intensive real-time monitoring represents what users actually want, nor that the quality improvements justify the additional attention costs.

### Questionable Framing and Positioning

The paper frames "cognitive oversight" as a paradigm shift and claims to challenge assumptions about AI autonomy. This positioning is not supported. Human-in-the-loop machine learning, active learning, interactive machine learning, and mixed-initiative systems have explored human-AI collaboration for decades. The paper does not engage meaningfully with this literature or explain what distinguishes cognitive oversight from these established paradigms beyond terminology.

The criticism of existing systems as opaque "black boxes" mischaracterizes both their capabilities (as discussed above) and their design rationale. Streamlined interaction models may reflect informed product decisions based on actual user preferences rather than technical limitations. The paper assumes that more interaction opportunities equal better design without empirical support for this assumption.

### Methodological Flaws, Limited Scope, and Missing Analyses

Sample sizes are inadequate. With 15 evaluation dimensions and small samples, multiple comparison problems and false positive risks are severe. The evaluation relies almost entirely on subjective Likert-scale ratings. Objective metrics are absent: no factual accuracy rates, no citation quality analysis, no systematic task completion efficiency measurement, no resource cost accounting.

Potential biases are uncontrolled. Participants proposed their own research questions (confirmation bias), knew which system was "new" (experimenter demand), and were not blinded (expectancy effects). No measures address these threats to validity. Ablation studies are completely absent. The contributions of individual components and design principles are not isolated. Whether reported improvements come from system architecture or simply from increased user engagement remains unknown.

Evaluation is restricted to expert users with research experience. Whether the system benefits non-experts is unknown. Task scope is narrow, focusing exclusively on academic research scenarios. Generalization to other contexts (business intelligence, market analysis, legal research) is unexplored. Resource efficiency is not analyzed. The system generates extensive API calls (5 queries × 20 URLs per cycle, multiple model instances) but provides no cost analysis.

**Questions:**

**Q1**: Given that ChatGPT provides stop functionality, Claude shows thinking processes, and Perplexity displays search queries in real-time, how do you characterize your system's novelty? What specific capabilities does Deep Cognition provide that cannot be achieved through existing commercial systems?

**Q2**: Why did your experimental design not include a control where baseline systems are used optimally—with multi-turn interaction? How would Deep Cognition compare against ChatGPT or Claude with unrestricted conversation?

**Q3**: Your comparison shows Deep Cognition with continuous monitoring outperforms baselines with limited interaction. How much of this difference is attributable to system architecture versus more user input? Could you equalize interaction time and feedback rounds across conditions?

**Q4**: Could you provide evidence that intensive real-time monitoring is what users want? What percentage of tasks benefit from continuous oversight versus asynchronous refinement?

**Q5**: What are the specific algorithmic contributions beyond integrating existing techniques? What innovations could not be replicated with existing frameworks and appropriate interfaces?

**Q6**: Could you report objective quality metrics (factual accuracy, citation quality, resource costs) and effort-adjusted performance? What are actual user time investments and cognitive load?

**Q7**: Why are ablation studies absent? What are the individual contributions of transparency, interruptibility, architecture, and user engagement?

**Q8**: How does "cognitive oversight" differ fundamentally from established human-in-the-loop concepts? What constitutes a genuinely new paradigm versus rebranding?

---

### Official Review · Reviewer_JFUQ · 2025-11-01

**Soundness:** 2
**Presentation:** 3
**Contribution:** 2
**Rating:** 4
**Confidence:** 3

**Summary:**

This paper introduces DeepCognition, a multi-agent framework that transform deep research tasks from asynchronous "input-wait-output" systems into real-time, human-AI collaborative cognitive systems. It allows users to view, interrupt, and guide the AI's reasoning process during complex research tasks lasting 15-30 minutes and involving hundreds of web resources.
DeepCognition is built through three technical pillars: transparent and
interruptible AI reasoning, fine-grained bidirectional dialogue, and a shared cognitive context. DeepCognition employs a layered state manager architecture, an option-driven clarification mechanism, a distributed sub-browsing agent cluster, and intermediate report generation capabilities.
Evaluation through user studies and benchmark tests (browsecomp-ZH, X-bench) shows improvements in both interaction quality and report quality when users actively interact with the system.  Furthermore, the benchmark test accuracy reached 72.73%, compared to Gemini/OpenAI: 40.91%, Grok 3: 22.73%.

**Strengths:**

1. Comprehensive  multi-agent system design .

2. Interesting and practical problem of interactive deep research system.

**Weaknesses:**

1. The paper compare between interactive Deep Cognition system with expert users actively guiding it versus non-interactive commercial baselines that never receive human intervention during research,
which is a "expert-guided system versus autonomous system" comparison. The 72.73% versus 40.91% accuracy difference may reflect the effect of both human expertise as much as architectural innovation.
More ablation study would be necessary for a fair comparison.
eg: (a) Deep Cognition in fully autonomous mode without any interaction affordances, not just "DC w/o Interaction" where it's unclear what is disabled. (b) Commercial baselines given equivalent human guidance budgets, e.g., same number of interventions


2. Small sample size of human study. Only at most 13 participants  is insufficient for drawing strong conclusions, particularly when most metrics are subjective assessments. The paper reports no confidence intervals, significance tests, p-values, or variance measures.

3. potential bias in human study. eg: All participants appear to be Chinese speakers since browsecomp-ZH is Chinese-focused, providing no evidence of cross-linguistic or cross-cultural generalization.

**Questions:**

Can you provide results for Deep Cognition in a fully autonomous mode (with all interaction mechanisms disabled, not just "w/o Interaction") on the same benchmarks, to establish whether the architecture itself provides value independent of human guidance?

---

### Official Review · Reviewer_cEid · 2025-11-10

**Soundness:** 2
**Presentation:** 2
**Contribution:** 2
**Rating:** 4
**Confidence:** 5

**Summary:**

This paper studies the deep research problem via the collaboration between humans and AI, e.g., large language models (LLMs). The author proposes a framework called Deep Cognition to embed real-time human expertise into the LLM reasoning process for complex research tasks by three principles, i.e., transparency, real-time intervention, and fine-grained interaction. The author argues that the main difference between Deep Cognition and the existing deep research projects from existing systems designed by OpenAI, Google, and Perplexity AI is that the user can pause the research progress and input feedback and requirements at any moment. Experiments are conducted in both an objective and a subjective manner, where the author (1) recruited 13 participants with prior research experience to use the proposed system and (2) benchmarked the system on 22 questions from browsecomp-ZH and the first 20 questions from xbench-deep research, showing the effectiveness of the proposed system.

**Strengths:**

1. The deep research task is challenging, and its advancement could be crucial for advancing LLM research. The author identifies a relevant issue with the existing research system and proposes a systematic solution to involve users in the research progress.

2. The proposed Deep Cognition can achieve better results than existing LLMs in the deep research task.

**Weaknesses:**

1. Missing critical technical details. As the main difference between the proposed Deep Cognition and the existing LLM is in the real-time intervention for the research progress and dynamically changing the planning for the LLM agent, the author should provide the detailed technical design in the main paper. However, most of Section 2 is only a high-level description, and it is hard to get the technical contribution and insight of the proposed system, e.g., Section 2.4 claimed that the author proposed "a dynamic research planning generation mechanism", but there is no detailed description of it in the main paper. The lack of technical details makes it hard to convince the reviewer of the technical contribution claimed by the author.

2. Evaluation metrics are unclear. As the deep research task is complicated, proper evaluation metrics are demanded to comprehensively evaluate the proposed system's capability. However, there is no description of the evaluation metrics for the proposed system in either the main paper or the Appendix. Appendix D should be a section for Evaluation Metrics Design, but it is empty, making it challenging for the reviewer to believe the reported experimental results in the main paper.

3. Experimental Setup may be biased. In Page 6, the author proposes an evaluation study (Study 2) that only selects representative questions from the browsecomp-ZH and xbench-deep research benchmarks, "given that the expert annotators are native Chinese speakers with domain expertise". Such an evaluation setup may introduce biases, and it may not be fair to other models, such as OpenAI or Gemini, given that they may not have been optimized specifically for the Chinese domain.

**Questions:**

Please refer to the Weaknesses section for the details.

---

> ### Author Response · Authors · 2025-11-27
>
> We sincerely thank the reviewer for their careful reading and insightful evaluation of our work. The key issues raised are highly valuable in helping us improve both the overall quality and the technical clarity of the paper.
>
> **Comment 1**: Missing critical technical details. As the main difference between the proposed Deep Cognition and the existing LLM is in the real-time intervention for the research progress and dynamically changing the planning for the LLM agent, the author should provide the detailed technical design in the main paper. However, most of Section 2 is only a high-level description, and it is hard to get the technical contribution and insight of the proposed system, e.g., Section 2.4 claimed that the author proposed "a dynamic research planning generation mechanism", but there is no detailed description of it in the main paper. The lack of technical details makes it hard to convince the reviewer of the technical contribution claimed by the author.
>
> **Response to comment 1**: We agree that the main text provided a high-level overview due to space constraints. we **significantly rewrite Section 2** to present our core system design and technical detail.
>
> **Comment 2**: Evaluation metrics are unclear. As the deep research task is complicated, proper evaluation metrics are demanded to comprehensively evaluate the proposed system's capability. However, there is no description of the evaluation metrics for the proposed system in either the main paper or the Appendix. Appendix D should be a section for Evaluation Metrics Design, but it is empty, making it challenging for the reviewer to believe the reported experimental results in the main paper.
>
> **Response to comment 2**: We sincerely apologize for the formatting confusion. While the header for Appendix D (Page 19) appears empty due to a layout error, the detailed definitions and scoring rubrics are present in Appendix. We now **move the metric definitions** into the **main text of Section 4.1** to ensure the evaluation criteria are visible before the results are presented.
>
> **Comment 3**: Experimental Setup may be biased. In Page 6, the author proposes an evaluation study (Study 2) that only selects representative questions from the browsecomp-ZH and xbench-deep research benchmarks, "given that the expert annotators are native Chinese speakers with domain expertise". Such an evaluation setup may introduce biases, and it may not be fair to other models, such as OpenAI or Gemini, given that they may not have been optimized specifically for the Chinese domain.
>
> **Response to comment 3**: We appreciate this concern regarding fair comparison. We have explicitly clarify that the core contribution of the Deep Cognition framework lies in its fully **transparent human–AI interaction design and multi-agent architecture**, rather than in the underlying large language models themselves. Therefore, any potential “language advantage” stems from our proposed interactive research architecture, not from the pretrained language capabilities of the base models. We believe the bias is minimal and the comparison remains valid for the following reasons:
>
> (1) **Model Capability**: The baselines compared (OpenAI GPT-4o, Gemini 1.5 Pro) are State-of-the-Art multilingual models that have demonstrated near-native proficiency in Chinese in public benchmarks (e.g., C-Eval, CMMLU). **They are not "disadvantaged" by the language; rather, they are the strongest available baselines for any language**.
>
> (2) **Task Universality**: The questions selected from browsecomp-ZH and xbench focus on reasoning and multi-hop information retrieval (e.g., "systematically understand perspectives on AI agent roles," Page 20), rather than culturally specific trivia. The complexity lies in the research process (planning, filtering, synthesizing), which is language-agnostic.
>
> (3) **Annotator Expertise**: We used native speakers specifically to ensure the highest quality of evaluation. Deep research reports are complex; evaluating them requires a nuance that non-native speakers might miss. Using native benchmarks ensured our experts could accurately judge "Hallucination" vs. "Nuanced Fact." To address this, we will add a subsection in the "limitations" discussing the language choice, Recruit an equal number of native English speakers to mitigate cultural differences(up sample from 13 to 26).

---

### Note · Authors · 2026-01-05

I have read and agree with the venue's withdrawal policy on behalf of myself and my co-authors.